# Cordycepin Resensitizes T24R2 Cisplatin-Resistant Human Bladder Cancer Cells to Cisplatin by Inactivating Ets-1 Dependent MDR1 Transcription

**DOI:** 10.3390/ijms21051710

**Published:** 2020-03-02

**Authors:** Sang-Seok Oh, Ki Won Lee, Hamadi Madhi, Jin-Woo Jeong, Soojong Park, Minju Kim, Yerin Lee, Hyun-Tak Han, Cheol Hwangbo, Jiyun Yoo, Kwang Dong Kim

**Affiliations:** 1Gene & Cell Therapy Team, Division of Drug Development & Optimization, New Drug Development Center, Osong Medical Innovation Foundation, Osongsaengmyung-ro 123, Osong-eup, Heungdeok-gu, Cheongju-si, Chungbuk 28160, Korea; ssoh@kbiohealth.kr; 2Division of Applied Life Science (BK21 Plus), Gyeongsang National University, Jinju 52828, Korea; leemaskup@naver.com (K.W.L.); soojongpark@kribb.re.kr (S.P.); kmj941226@naver.com (M.K.); renii21@naver.com (Y.L.); chwangbo@gnu.ac.kr (C.H.); yooj@gnu.ac.kr (J.Y.); 3Freshwater Bioresources Utilization Bureau, Nakdonggang National Institute of Biological Resources, Sangju 37242, Korea; jwjeong@nnibr.re.kr; 4PMBBRC, Gyeongsang National University, Jinju 52828, Korea; entreluzyluz@naver.com; 5Division of Life Science, Gyeongsang National University, Jinju 52828, Korea

**Keywords:** cordycepin, cisplatin-resistance, resensitization, MDR1, Ets-1, PI3K

## Abstract

Tumor cell resistance to anti-cancer drugs is a major obstacle in tumor therapy. In this study, we investigated the mechanism of cordycepin-mediated resensitization to cisplatin in T24R2 cells, a T24-derived cell line. Treatment with cordycepin or cisplatin (2 μg/mL) alone failed to induce cell death in T24R2 cells, but combination treatment with these drugs significantly induced apoptosis through mitochondrial pathways, including depolarization of mitochondrial membranes, decrease in anti-apoptotic proteins Bcl-2, Bcl-xL, and Mcl-1, and increase in pro-apoptotic proteins Bak and Bax. High expression levels of MDR1 were the cause of cisplatin resistance in T24R2 cells, and cordycepin significantly reduced MDR1 expression through inhibition of MDR1 promoter activity. MDR1 promoter activity was dependent on transcription factor Ets-1 in T24R2 cells. Although correlation exists between MDR1 and Ets-1 expression in bladder cancer patients, active Ets-1, Thr38 phosphorylated form (pThr38), was critical to induce MDR1 expression. Cordycepin decreased pThr-38 Ets-1 levels and reduced MDR1 transcription, probably through its effects on PI3K signaling, inducing the resensitization of T24R2 cells to cisplatin. The results suggest that cordycepin effectively resensitizes cisplatin-resistant bladder cancer cells to cisplatin, thus serving as a potential strategy for treatment of cancer in patients with resistance to anti-cancer drugs.

## 1. Introduction

Although chemotherapy is thus far the most effective cancer treatment, the clinical response of a patient to this treatment varies widely, and survival is often unsatisfactory. Resistance to anti-cancer drugs often hampers the therapeutic effects of chemotherapy. Although conventional anti-cancer drugs have different targets and mechanisms of action, most of them kill cancer cells by inducing apoptosis. However, due to the imbalanced regulation of apoptotic machinery in cancer cells, a small subset of them can acquire resistance to anti-cancer drugs [1].

Resistance to anti-cancer drugs is conferred by a multitude of mechanisms, including decreased drug influx [2], increased drug efflux [3], and activation of DNA repair mechanisms [4]. The best studied mechanism of drug resistance in tumor cells is an increased efflux of anti-cancer drugs. This mechanism is very effective in preserving intracellular drug concentrations below the apoptosis-triggering threshold by using a variety of ATP-dependent active drug transporters, such as multidrug resistance protein 1 (MDR1), breast cancer resistant protein (BCRP), and multi-drug resistance-associated protein 1 (MRP1) [5,6]. These transmembrane active transporters play an important role in not only recognizing anti-cancer drugs, but also in detoxifying intracellular compartments by pumping these drugs out. Novel pharmacological approaches to overcome drug resistance in cancer have been established in recent decades. Targeting active drug transporters such as MDR1, for example, can resensitize drug-resistant tumor cells to anti-cancer drugs [7,8].

Cordycepin (3′-deoxyadenosine) is a major bioactive substance extracted from *Cordyceps militaris* mushrooms—a traditional Chinese medicine [9,10]. Cordycepin exhibits anti-tumor qualities, including anti-angiogenic, anti-metastatic, anti-proliferative, and pro-apoptotic activity in cancer cells [11,12,13].

In this study, we investigated the mechanism of cordycepin-mediated resensitization to cisplatin in T24R2 cells, a cisplatin-resistant cell line derived from the T24 human bladder cancer cell line [14], suggesting that cordycepin may be developed as a candidate for combination therapy combinations in patients with cisplatin resistance.

## 2. Results

### 2.1. Cordycepin Resensitized T24R2 Cells to Cisplatin

The MTT assay was used to confirm the resistance of T24R2 cells to cisplatin. Cell viability was quantified 24 h after cisplatin treatment of T24 and T24R2 cells at concentrations of 1 or 2 μg/mL. Although cisplatin induced concentration-dependent T24 cell death, no significant effect was observed in T24R2 cells, which showed clear resistance to cisplatn (Figure 1A). To investigate the effect of cordycepin on T24R2 cells, we treated T24R2 cells with various concentrations of cordycepin alone or with a combination of cordycepin and cisplatin, and measured cell viability using the MTT assay (Figure 1B). While cordycepin-induced cytotoxicity in T24R2 cells was slightly increased at a high dose of cordycepin (50 μg/mL), combination treatment with cordycepin and cisplatin significantly induced cell death starting at 20 μg/mL of cordycepin. Cytotoxicity caused by apoptosis was confirmed by propidium iodide sub-G_1/0_ (Figure 1C) and TUNEL assays (Figure 1D). These data suggest that cordycepin resensitizes T24R2 cells to cisplatin.

### 2.2. Cordycepin-Mediated Resensitization of T24R2 Cells to Cisplatin is Induced by Apoptosis via the Mitochondrial Pathway

To investigate the mechanism of cordycepin-mediated resensitization of T24R2 cells to cisplatin, we assessed protein expression levels involved in caspase pathway activation. In brief, T24R2 cells were treated with cisplatin (2 μg/mL) and cordycepin (20 μg/mL) for 24 h, and subsequently the levels of caspase-3, -9, and poly-ADP-ribose polymerase (PARP) were evaluated by Western blotting. Interestingly, the combination treatment with cisplatin and cordycepin induced cleavage of caspase-3, -9, and PARP (Figure 2A). As shown in Figure 2B, while the levels of anti-apoptotic proteins Mcl-1, Bcl-2, and Bcl-xL were significantly reduced, the levels of pro-apoptotic proteins Bax and Bak were considerably increased in the T24R2 cells co-treated with cisplatin and cordycepin. Moreover, to confirm that the combination of cisplatin and cordycepin activated the caspase cascade in T24R2 cells via the mitochondrial pathway, we examined active Bax redistribution. Combination treatment induced active Bax translocation to the mitochondrial membrane as shown in Figure 2C. In addition, the combination treatment induced a significant loss of the mitochondrial potential, which is considered a hallmark of apoptosis (Figure 2D). These data clearly demonstrate that cordycepin combined with cisplatin induces apoptosis in T24R2 cells via the mitochondrial pathway-dependent caspase activation cascade.

### 2.3. Reduction of MDR1 Expression is Involved in Cordycepin-Mediated Resenstization of T24R2 Cells

MDR1 expression was increased in T24R2 cells compared with T24 cells (Figure 3A) and was downregulated by cordycepin at the transcriptional level (Figure 3B,C). Moreover, to confirm that MDR1 is directly associated with T24R2 cell resistance to cisplatin, we knocked down expression of MDR1 in T24R2 cells. After treatment with 2 μg/mL cisplatin for 24 h, siMDR1-transfected T24R2 cells showed a much higher cell death rate compared with siGFP-transfected T24R2 cells (Figure 3D). Furthermore, MDR1 knockdown resulted in reduction of the anti-apoptotic proteins Bcl-2 and Mcl-1, as well as in induction of the pro-apoptotic proteins Bax and Bak (Figure 3E). To confirm whether cordycepin inhibits drug efflux mediated by a decrease of MDR1 expression, we assayed the intracellular levels of rhodamine in T24, T24R2, cordycepin-treated T24R2 cells, and siMDR1-treated T24R2 cells. T24R2 cells exhibited very low levels of cytosolic rhodamine accumulation, whereas T24 cells contained high levels of rhodamine. As expected, cordycepin-treated T24R2 and siMDR1-treated T24R2 cells contained high levels of rhodamine (Figure 3F). Taken together, these data suggest that cordycepin-mediated reduction of MDR1 is a major mechanism inducing resensitization of T24R2 cells to cisplatin.

### 2.4. Cordycepin Regulates MDR1 Promoter Activity

Because the level of MDR1 mRNA was significantly decreased in cordycepin-treated T24R2 cells (Figure 3B), we investigated whether cordycepin suppresses MDR1 expression by downregulating MDR1 promoter activity. Using a dual luciferase reporter gene assay, the activity of the MDR1 promoter (from −751 to +122 bp) was compared in T24 and T24R2 cells. As shown in Figure 4A, T24R2 cells showed higher luciferase activity than T24 cells. Furthermore, cordycepin treatment reduced MDR1 promoter activity in T24R2 cells to the same level as in T24 cells (Figure 4B).

A previous report suggested that Ets-1 activates the human MDR1 promoter in the human osteosarcoma cell line Saos-2 [15]. To investigate whether Ets-1 is necessary for MDR1 expression in T24R2 cells, we constructed Ets-1 mt-1, Ets-1 mt-2, and Ets-1 mt-1 & 2 with mutations in the Ets-1 binding sequence of the MDR1 promoter; all of these mutants lacked promoter activity in T24R2 cells (Figure 4C). Thus, Ets-1 may be necessary for MDR1 promoter activation in T24R2 cells. To confirm whether Ets-1 binds to the MDR1 promoter in T24R2 cells, and, if so, whether the binding is inhibited by cordycepin treatment, we performed ChiP assay with an anti-Ets-1 antibody. While Ets-1 did not bind to the promoter in T24 cells, it bound directly to the MDR1 promoter in T24R2 cells. Cordycepin effectively inhibited the binding of Ets-1 to the MDR1 promoter (Figure 4D). Therefore, we suggest that cordycepin downregulates MDR1 expression via inhibition of transcription factor activity of Ets-1, which sensitizes T24R2 cells to cisplatin.

### 2.5. Cordycepin Inhibits PI3K/AKT Activation that Phosphorylates Ets-1

To determine whether a correlation exists between the expression of Ets-1 and that of MDR1, the cBioPortal database for cancer genomics [16] was utilized in January 2018. The mRNA expression levels of Ets-1 and MDR1 from 413 bladder cancer patients demonstrated positive correlation (Figure 5A). Although cordycepin reduced MDR1 expression at the transcription level (Figure 3B,C), transcript and protein levels of Ets-1 were unaffected by cordycepin (Figure 5B). These data suggest that cordycepin may reduce MDR1 expression by inhibiting the function, but not the expression, of Ets-1. Accordingly, we investigated post-translational modifications of Ets-1 affected by cordycepin. Phosphorylation of Ets-1 at Thr38 induces its activation as a transcription factor [17]. As shown in Figure 5C, cordycepin treatment decreased the level of phospho-Ets-1 (Thr38), which was followed by a reduction of MDR1 expression. Although the levels of activated ERK and AKT were higher in T24R2 cells than those in T24 cells, cordycepin treatment had little effect on ERK activation, but significantly reduced AKT activation in T24R2 cells (Figure 5D, Appendix A). Inhibition of PI3K by wortmannin, a specific inhibitor of PI3K, also attenuated Ets-1 phosphorylation and reduced MDR1 expression as observed in cordycepin-treated T24R2 cells (Figure 5E). To confirm whether PI3K inhibition confers improved sensitivity to cisplatin in T24R2 cells, these cells were treated with cordycepin or wortmannin in the presence or absence of cisplatin. Wortmannin treatment resensitized T24R2 cells to cisplatin in a manner similar to that of cordycepin treatment (Figure 5F). These data suggest that the inhibition of PI3K activation by cordycepin attenuates Ets-1 activity, which leads to resensitization of T24R2 cells to cisplatin through the reduction of phospho-Ets-1.

## 3. Discussion

Although chemotherapy is one of the most effective methods of tumor therapy, tumor cell acquisition of drug resistance is one of the major obstacles for efficacious chemotherapy. Therefore, the development of strategies to overcome anti-cancer drug resistance is imperative. One of these strategies is combination chemotherapy, and drug selection for the establishment of combination therapy strategies is very important. MDR1 (P-glycoprotein), a well-known ATP-binding cassette (ABC) transporter, is integrated into the plasma membrane and reduces the efficacy of chemotherapy by shortening the retention time of intracellular drugs. Indeed, elevated expression of MDR1 induces drug resistance in many tumors. Thus, inhibition of MDR1 is a potential strategy for resensitizing anti-cancer drug-resistant cancer cells to applicable drugs [18,19]. Although several pharmacological inhibitors against MDR1 have been proposed to overcome tumor drug resistance, their toxicities are generally an obstacle to clinical application [20,21].

In this study, we found that cordycepin, a compound isolated from *Cordyceps militaris*, attenuated Ets-1 activity through inhibition of PI3K. This played an important role in increasing the sensitivity of T24R2 cells to cisplatin by reducing the promoter activity of MDR1. Ets-1 is an essential transcription factor regulating the transcription of MDR1 and promoting anti-cancer drug resistance in various tumors [15,22]. MDR1 promoter activity was much higher in T24R2 cells than in T24 cells, and cordycepin treatment inhibited activation of the MDR1 promoter. This activity was also eliminated by mutation of the Ets-1 binding sites on the MDR1 promoter, as Ets-1 was unable to bind the MDR1 promoter mutants in cordycepin-treated T24R2 cells (Figure 4). These data suggest that cordycepin inhibits MDR1 expression through inhibition of Ets-1 activity.

Transcription and post-translational modification regulate Ets-1 expression and its activity [23,24]. MDR1 expression showed a quantitative linear correlation with Ets-1 expression at the transcriptional level in human bladder cancer patients according to the analysis using cBioPortal for cancer genomics [16]. Although T24 and T24R2 cells showed similar Ets-1 expression levels of mRNA and proteins, MDR1 expression was higher in T24R2 cells than in T24 cells. These data suggest that cordycepin-mediated reduction of MDR1 expression may not be dependent on the amount of Ets-1 expression. Several reports have demonstrated that the MEK/ERK signaling pathway regulates Ets-1 expression and its activity through phosphorylation of Ets-1 at Thr38 [17,25,26,27,28]. Some reports suggested that cordycepin could regulate ERK activation in murine cells associated with osteoblast and osteoclast, adipocytes and human hepatocarcinoma cells [29,30,31]. Strangely, a decrease in ERK activation was unclear or little in T24R2 cells treated with cordycepin, whereas the inhibition of AKT activation was observed. Previous studies have reported that AKT activation induces Ets1 expression [32,33,34]. In our study, no decrease of Ets-1 expression was observed, whereas phospho-Ets-1 (Thr38) significantly decreased in T24R2 cells treated with cordycepin or wortmannin. These data suggest that PI3K inhibition mediated by cordycepin may inhibit the phosphorylation of Ets-1 and MDR1 expression in T24R2 cells and resensitize the cells to cisplatin (Figure 5). However, Scansite 3.0 motif analysis showed that Thr38 at Ets-1 would not be phosphorylated by any kinase, AKT, PDK, or PKC, directly associated with PI3K activation. A recent study suggested that direct inhibition of PI3K using the specific inhibitor BAY-1082439 or knockout of PI3K 110α or 110β downregulates MDR1 expression independently of the AKT pathway in non-small cell lung cancer (NSCLC) [35]. Precise details of PI3K-mediated regulation of Ets-1 expression, however, remain unclear.

In this report, we demonstrate a new aspect of cordycepin on overcoming drug resistance in the cisplatin-resistant bladder cancer T24R2 cell line. We investigated the mechanism by which MDR1 expression is inhibited through regulation of Ets-1 activity by cordycepin. We suggest that cordycepin inhibits activation of Ets-1 through inhibition of the PI3K pathway. Although further studies are required to address how the PI3K pathway regulates phosphorylation of Ets-1, cordycepin may be a potential candidate for combination therapy with cisplatin in the treatment of patients exhibiting cisplatin resistance.

## 4. Material and Methods

### 4.1. Cell lines and Cultures

The T24 cell line was purchased from the American Type Culture Collection (Manassas, VA, USA). The T24R2 cell line, a cisplatin-resistant derivative cell line of T24, was established through serial desensitization of T24 cells and showed resistance to 2 μg/mL cisplatin [14]. The cells were cultured as monolayers in Dulbecco’s modified Eagle’s medium (DMEM; Sigma, St. Louis, MO, USA) supplemented with 10% (*v*/*v*) heat-inactivated fetal bovine serum (Sigma-Aldrich) and 1% penicillin/streptomycin (Lonza, Basel, Switzerland); they were maintained at 37 °C in a humidified chamber with 5% CO_2_.

### 4.2. Antibodies and Reagents

Cordycepin purified from *Cordyceps militaris* was purchased from Sigma, and cisplatin was purchased from ILDONG Pharmaceutical (Korea). Anti-Bcl-2, anti-Bax, anti-Bak, anti-MDR1, anti-Mcl-1, anti-caspase-3, anti-caspase-9, anti-cleaved poly-ADP-ribose polymerase (PARP), anti-phospho-AKT (Thr308), and anti-AKT antibodies were purchased from Cell Signaling Technology (Danvers, MA, USA). Anti-α-tubulin, anti-Bcl-xL, and anti-Ets-1 (C-4) antibodies were purchased from Santa Cruz Biotechnology (Dallas, TX, USA). Anti-pEts-1 (T38) antibody was purchased from Invitrogen (Carlsbad, CA, USA). Horseradish peroxidase-conjugated goat anti-mouse IgG, goat anti-rabbit-IgG, 3-(4,5-Dimethyl-2-thiazolyl)-2,5-diphenyl-2H-tetrazolium bromide (MTT), propidium iodide (PI), cordycepin, and 4′, 6-diamidino-2-phenylindole (DAPI) were purchased from Sigma-Aldrich.

### 4.3. MTT Assay

An MTT assay was performed on the T24 and T24R2 cells to determine cell viability. Cells were plated in a 96-well plate (5 × 10^3^ cells/well) and incubated for 24 h in the presence or absence of cisplatin and/or cordycepin. MTT solution (5 mg/mL) was added to the wells, and the cells were incubated for another 3 h. Following incubation, the medium was removed, and 200 μL DMSO was added to each well to extract the formazan products produced by viable cells. The absorbance of the solutions was measured on a Bio-Rad 550 microplate reader at 595 nm. Relative cell viability (%) was determined by comparing the absorbance at 595 nm with control, which was treated with DMSO.

### 4.4. PI Staining

The cells were fixed with 70% ethanol and stored at −20 °C overnight. The cells were washed with PBS (phosphate buffer saline), treated with RNase A (ribonuclease A, Sigma), and then stained with PI (Sigma). The cells were then subjected to flow cytometric analysis using FACSVerse^TM^ (Becton, Dickinson and Company, Franklin Lakes, NJ, USA) and the data were analyzed using the FlowJo software (FLOW JO LLC, USA).

### 4.5. RNA Isolation and RT-PCR Analysis

Total RNA was extracted from cells using the total RNA isolation solution (RiboEx^TM^; GeneAll, Seoul, Korea) according to the manufacturer′s instructions. RNA was quantified using the NanoDrop^TM^ spectrophotometer (ThermoFisher Scientific, Waltham, MA, USA). cDNA (complementary DNA) was synthesized with oligo-dT (oligo-deoxythymidine) primers and reverse transcriptase (Fermentas, Sankt Leon-Rot, Germany). Reverse-transcriptase polymerase chain reaction (RT-PCR) was performed using a PCR detection kit (Solgent Co. LTD, Daejeon, Korea). The following primers were used for amplification: GAPDH, forward 5′-CCATCACCATCTTCCAGGAG-3′, and reverse 5′-ACAGTCTTCTGGGTGGCAGT-3′; MDR1, forward 5′-GACACCACTGGAGGGTGACT-3′, and reverse 5′-GGCGTTTGGAGTGGTAGAAA-3′; and Ets-1 forward 5′-ACCCAGATGAGGTGGCCAGG-3′, and reverse 5′-TCAGGGGTGTACCCCAGCAG-3′.

### 4.6. Immunoblotting Analysis

Total proteins extracted from cells were quantified using the Bradford method. Equal amounts of proteins were separated using 6%−15% SDS-PAGE (sodium dodecyl sulfate–polyacrylamide gel electrophoresis). After cold transfer onto a PVDF (polyvinylidene difluoride) membrane, nonspecific binding sites were blocked for 1 h with 5% skim milk (Difco Laboratories, Surrey, UK) in Tris-buffered saline containing 0.1% Tween 20. The membranes were treated with primary antibodies at 4 °C overnight, and HRP (horseradish peroxidase)-conjugated secondary antibodies were incubated with the membranes at room temperature for 1 h. HRP signal was visualized using Clarity™ ECL Western Blotting Substrate (Bio-Rad, Hercules, CA, USA).

### 4.7. Confocal Microscopy

T24R2 cells were seeded and incubated overnight onto 25 mm round glass coverslips. Following induction of apoptosis by a co-treatment with cordycepin and cisplatin, the cells were fixed with 4% paraformaldehyde. Fixed cells were treated with an active Bax antibody (1:50 in PBS) overnight at 4 °C, and then with α-Mouse-FITC (1:100 in PBS) for 2 h at room temperature. Coverslips were mounted on glass slides with a mounting solution, and fluorescence was visualized with an Olympus FluoView™ FV1000 Confocal Microscope (Olympus, Tokyo, Japan). For terminal deoxynucleotidyl transferase (TdT)-mediated biotinylated UTP (uridine 5’-triphosphate) nick end labeling (TUNEL) analysis, T24R2 cells were treated with cisplatin (2 μg/mL), cordycepin (30 μg/mL) or a combination of the two for 24 h. DNA fragmentation was determined by a TUNEL staining kit (Promega, Madison, WI, USA) according to the manufacturer’s instructions.

### 4.8. Rhodamine123 Uptake/Retention Assay

Rhodamine123 (Rh123) uptake/retention experiments were performed by adding 0.25 µM of Rh123 following cell harvest. After 1 h of incubation in the presence of Rh123, cells were washed with PBS two times. Cellular efflux of Rh123 was measured using flow cytometry (FACSVerse^TM^).

### 4.9. Construction of Luciferase Reporter Plasmid and Dual Luciferase Assay

The 873 bp fragment (from −751 to +122) was excised from a pMDR1-1202 cloning recombinant vector purchased from Addgene (Cambridge, MA, USA) and ligated into the equivalent site of a pGL3-basic vector (Promega) to form the pGL3-MDR1 construct. Site-directed mutagenesis was performed using a KOD-Plus-Mutagenesis kit (TOYOBO, Osaka, Japan) with a template pGL3-MDR1. The primers used for the different mutation were as follows: Ets-1 mt-1, forward 5′- TTAACAGCGCCGGGGCGTGGGCTGA-3′ and reverse 5′- TGCCCAGCCAATCAGCCTCACCACA-3′, and Ets-1 mt-2, forward 5′-TTAAGC CTGAGCTCATTCGAGTAGC-3′ and reverse 5′-TGTGGCAAAGAGAGCGAAGCGGCTG-3′. T24R2 cells were grown at 37 °C in 5% CO_2_ and transfected with FuGENE^TM^ HD in 24-well plates with 7 × 10^4^ cells. All the cells were analyzed for dual-luciferase reporter gene expression 48 h after completion of the transfection procedure. Activities of firefly luciferase and *Renilla* luciferase in the pGL3-MDR1 vector were determined following the dual luciferase reporter assay protocol recommended by Promega.

### 4.10. Chromatin Immunoprecipitation (ChIP) Assay

T24, T24R2, and cordycepin-treated T24R2 cells were fixed with 1% formaldehyde solution. Chromatins were prepared from the fixed cells and fragmented by sonication. Immunoprecipitation of cross-linked protein/DNA was performed by using the Ets-1 (C-4) mouse monoclonal antibody. Briefly, 2 μg of the Ets-1 antibody was bound to chromatin mixture at 4 °C overnight with rotation, and then 30 μL of protein A magnetic beads (Merck Millipore) were added and additionally incubated at 4 °C overnight with rotation. Then real-time PCR analysis using precipitated DNA was performed. The following primers were used for amplification: MDR1 promoter, forward 5′-CGTTTCTCTACT TGCCCTTTCT-3′ and reverse 5′-CCGGATTGACTGAATGCTGAT-3′.

### 4.11. Statistical Analysis

The data are expressed as the means ± SD from three independent experiments. Differences among means were analyzed using a one-way ANOVA or student’s *t*-test. *p* < 0.05 was considered to be significantly different.

## Figures and Tables

**Figure 1 ijms-21-01710-f001:**
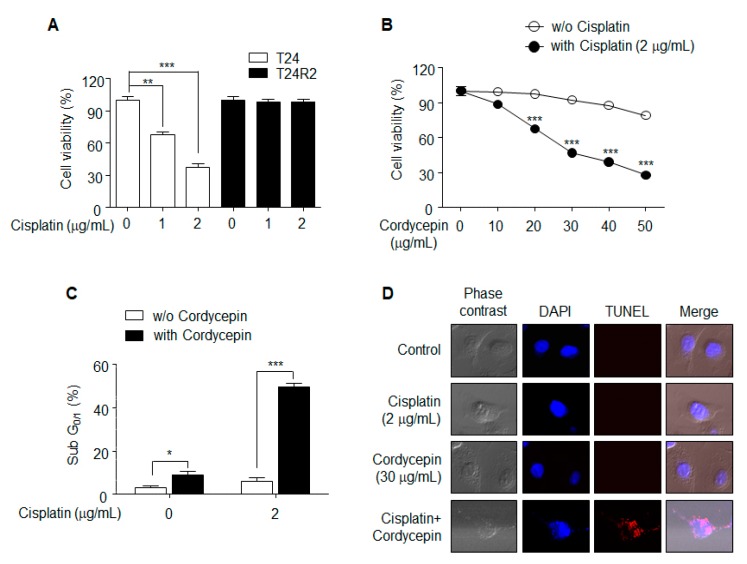
Effects of cordycepin treatment on T24R2 cell sensitivity to cisplatin. (**A**) T24 and T24R2 cells were incubated for 24 h with different concentrations of cisplatin (0, 1, or 2 μg/mL). Cell viability was determined by MTT (3-[4,5-dimethylthiazole-2-yl]-2,5-diphenyltetrazolium bromide) assay. (**B**,**C**) Cordycepin-mediated resensitization of T24R2 cells to cisplatin. T24R2 cells were treated with cordycepin in the presence or absence of cisplatin, and their viability was measured using the MTT assay (**B**). Bonferroni post hoc correction for multiple comparisons was performed to compare means by row (the effect of cordycepin was compared in matched group in the presence or absence of cisplatin). Determination of sub-G_1/0_ was accomplished using propidium iodide staining. (**C**,**D**) Apoptosis of T24R2 cells induced by combination treatment with 30 μg/mL cordycepin and 2 μg/mL cisplatin was analyzed by TUNEL (terminal deoxynucleotidyl transferase dUTP nick end labeling) and DAPI (4′,6-diamidino-2-phenylindole) staining. The results are representative of at least two independent experiments. * *p* < 0.05; ** *p* < 0.01; *** *p* < 0.001 by *t*-test.

**Figure 2 ijms-21-01710-f002:**
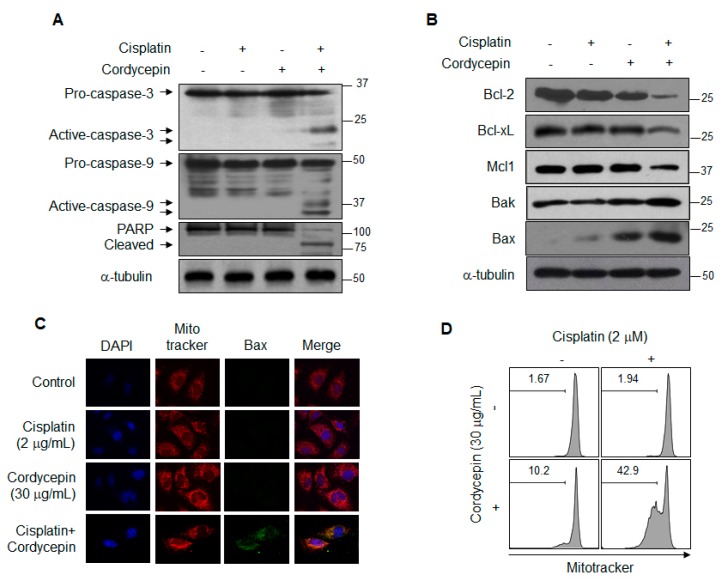
Induction of cordycepin-mediated apoptosis via the mitochondrial pathway in cisplatin-treated T24R2 cells. T24R2 cells were treated with 30 μg/mL cordycepin and/or 2 μg/mL cisplatin for 24 h, and whole cell extracts were analyzed by Western blotting using the indicated antibody against caspase-3, -9, or PARP cleavage (**A**), anti-apoptotic or pro-apoptotic proteins (**B**). α-tubulin was used as a loading control. T24R2 cells treated with cisplatin and/or cordycepin were immunostained with anti-active Bax and analyzed using a confocal microscope. (**C**) Active Bax was labeled with an FITC (fluorescein)-conjugated secondary antibody (green), and the mitochondria and nuclei were stained with MitoTracker CMXRos (red) and DAPI (blue), respectively. (**D**) Cells were stained with MitoTracker Red CMXRos, and mitochondrial depolarization was analyzed by flow cytometry. The results are representative of at least two independent experiments.

**Figure 3 ijms-21-01710-f003:**
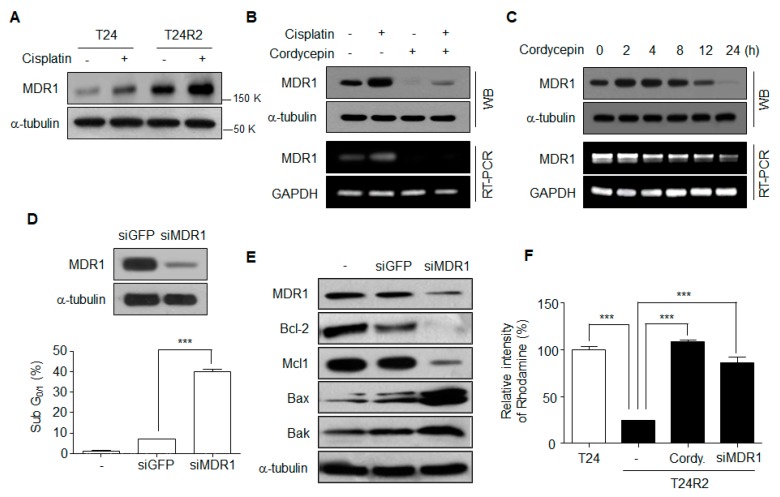
Cordycepin-mediated downregulation of MDR1 is associated with the T24R2 cell resistance to cisplatin. (**A**) Expression of multidrug resistance protein 1 (MDR1) in T24 and T24R2 cells. (**B**) MDR1 mRNA and protein levels in T24R2 cells treated with cordycepin and/or cisplatin. GAPDH, glyceraldehyde 3-phosphate dehydrogenase; WB, western blotting; RT-PCR, Reverse transcription ploymerase chain reaction. (**C**) MDR1 expression levels were analyzed after cordycepin treatment for the indicated times. (**D**) T24R2 cells were transfected with control siGFP or siMDR1 to knock down MDR1 expression. Representative flow cytometric analysis of DNA content in T24R2 cells treated with cisplatin under MDR1 knockdown (siRNA MDR1). (**E**) After transfection with MDR1 siRNA, T24R2 cells were treated for 24 h with 2 μg/mL of cisplatin. Anti-apoptotic proteins (Mcl-1 and Bcl-2) and pro-apoptotic proteins (Bax and Bak) were assessed by Western blotting. (**F**) Cells were transfected with siMDR1 and then treated with cordycepin (30 µg/mL) for 24 h. Cells were incubated for 1 h in the presence of Rhodamine123. The amount of intracellular Rhodamine123 was analyzed using flow cytometry, and the mean fluorescence intensity (MFI) was determined. The results are representative of at least two independent experiments. *** *p* < 0.001 by *t*-test.

**Figure 4 ijms-21-01710-f004:**
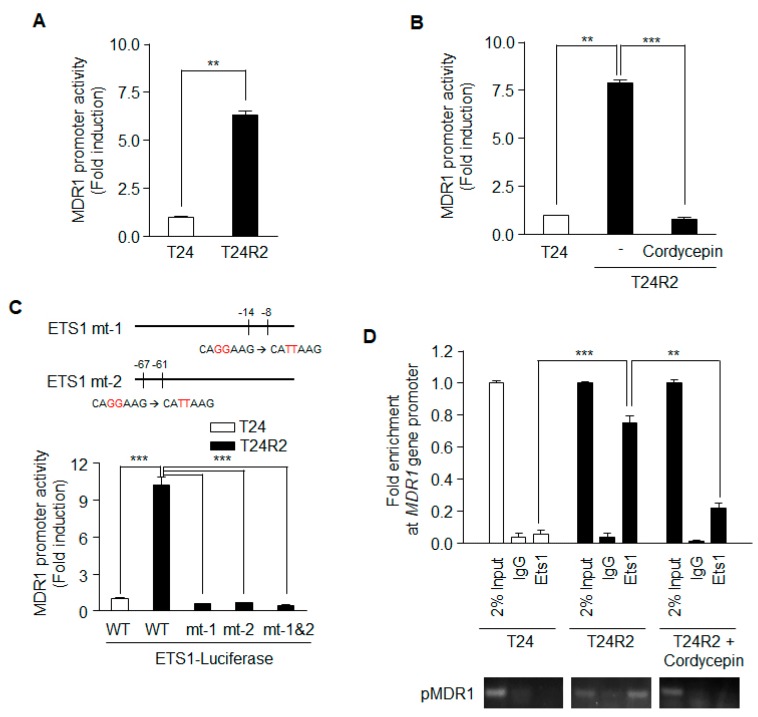
Ets-1 activity on the MDR1 promoter in T24R2 cells is attenuated by cordycepin treatment. T24 and T24R2 cells were transiently transfected with a pGL3-MDR1 luciferase vector that contained the 5’-promoter region of the MDR1 gene (873 bps: from −751 to +122), and luciferase activity was determined from on the basis of the cell lysates treated without (**A**) or with (**B**) cordycepin. (**C**) Site-directed mutagenesis was performed at Ets-1 biding sites (mt-1, mt-2, or mt-1 & 2) on the MDR1 promoter. pGL3 containing the wild-type (WT) MDR1 promoter or one of the site-mutated promoters was transfected into T24 or T24R2 cells, and luciferase activity was determined and normalized to the *Renilla* luciferase activity. (**D**) Using genomic DNA from T24, T24R2, or cordycepin-treated T24R2 cells, we performed chromatin immunoprecipitation (ChIP)-PCR. The results are representative of at least two independent experiments. ** *p* < 0.01, *** *p* < 0.001 by t-test.

**Figure 5 ijms-21-01710-f005:**
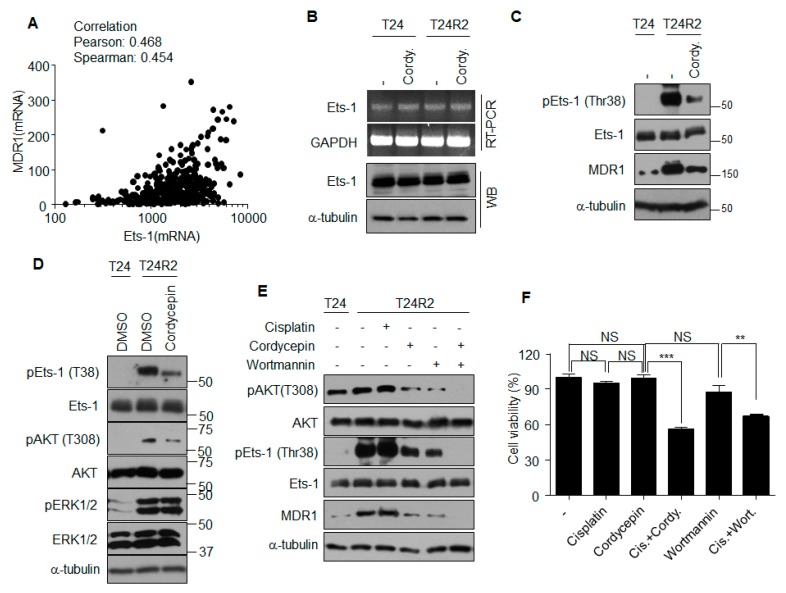
Cordycepin reduces Ets-1 phosphorylation through inhibition of the PI3K pathway, followed by reduction of MDR1 expression. (**A**) Correlation between Ets-1 and MDR1 mRNA expression levels was determined using the cBioPortal [16]. Pearson correlation = 0.468; Spearman correlation = 0.454. (**B**) T24 or T24R2 cells were treated with cordycepin for 24 h, and Ets-1 mRNA or protein levels were determined. (**C**) Cordycepin-mediated reduction of Ets-1 phosphorylation (pEst-1). Whole lysates from indicated experimental groups were analyzed by Western blotting using indicated antibodies. (**D**) Cordycepin inhibits active phosphorylation of AKT. (**E**) T24R2 cells were treated with cordycepin and/or wortmannin, and lysates were analyzed by Western blotting (WB) to determine correlation among active phosphorylation of AKT (T308, pAKT) or Ets-1 (Thr38) and expression of MDR1. Phosphorylation of ERK1/2, (pERK1/2). (**F**) T24R2 cells were treated with indicated agents for 24 h, and cell viability was determined by propidium iodide staining and flow cytometry analysis. The results are representative of at least two independent experiments. ** *p* < 0.01, *** *p* < 0.001 by t-test.

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
