# Peer review of "Cordycepin Resensitizes T24R2 Cisplatin-Resistant Human Bladder Cancer Cells to Cisplatin by Inactivating Ets-1 Dependent MDR1 Transcription"

_ijms, 2020, doi:10.3390/ijms21051710_

Round 1

Reviewer 1 Report

This is an interesting paper because it shows a potential mechanism for the synergism between cordycepin and chemotherapy drugs.

Key issues:

Were replicates of the westerns performed? This is necessary and should be indicated in the legend. It is now good practice to show replicates of western blots to the editor and/or the reviewer. I prefer to publish them with the paper in a supplement, but I will leave this to the journal to decide. Quantitation of westerns often places the bar so high that it prevents publication of useful results and I don’t encourage it. P13-14 : Figure 4 is a key figure with several issues. Firstly, the figure is inadequately described in the text. All results need a mention. Eg no mention is made in the text on the (lack of) effects on ERK signalling. It appears that total ERK and pERK are somewhat reduced, but the mis-shapen lane in for pERK makes it hard to be sure. A better blot is necessary to make this point. Secondly, it appears from the band shape that the total and phospho-ERK data were not taken from the same blot. There is no loading control for either blot. A loading control should be shown for every blot. Thirdly, as mentioned in the discussion, the pETS1Thr38 site is not a normal Akt target site, but thought to be controlled by ERK. In this light, the data in Figure 5D really need to be improved and westerns for ERK and pERK should also be shown for the wortmannin experiment, as crosstalk between the PI3K and ERK pathways could be occuring. It would be desirable to use a specific Akt inhibitor and MEK inhibitor to confirm that pETS1Thr38 is downstream of Akt and not ERK in these cells.  

Minor issues:

It is unclear to me which comparisons are tested in Figure 1B. Is untreated compared to the increased cordycepin concentration or is the difference between with and without cisplatin tested? Was a correction for multiple measurements, eg Dunnett or Bonferroni used?

P13 “Taken together, inhibition of AKT activation by cordycepin attenuates Ets-1 activity, which leads to resensitization of T24R2 to cisplatin through the reduction of pEts-1.” And p 15 “Therefore, cordycepin inhibits MDR1 expression via inhibition of Ets-1 activity.” P16 “Active PI3K/AKT pathway induces transcriptional upregulation of MDR1 through activation of Ets-1”. These statements are far too definitive. The data presented in the paper are only correlative. A definitive experiment would be for instance to induce resistance to the cisplatin-cordycepin combination by introducing a constitutively active ETS1 mutant and show upregulation of MDR1. If such an experiment cannot be provided, the statements should be rendered more tentative, eg “ These data suggest that..”

P15 “cordycepin inhibited MDR1 expression at the transcriptional level” To make this statement, there should also be measurement of endogenous MDR1 mRNA levels at the very least and preferably also of unspliced or chromatin bound MDR1 mRNA. This should be not too hard to do with qPCR. The methods suggest that qPCR for MDR1 mRNA was performed, but I don’t see the data.

P16 “Various growth factors, …, increase the mRNA level of Ets-1 through the Ras/Raf/MEK/ERK1/2 pathway” and “ Among the modifications of Ets-1, Thr-38 phosphorylation of Est-1 by ERK1/2 was well defined [27-29]. Strangely, no significant decrease in ERK activation was observed in T24R2 treated with cordycepin, while a significant inhibition in PI3K-mediated AKT activation was observed. PI3K/AKT inhibition using wortmannin significantly inhibited the phosphorylation of Ets-1 and MDR1 expression in T24R2 (Figure 5E), and also resensitized T24R2 to cisplatin (Figure 5F). Although it was known that AKT can affect Ets-1 expression levels [30], cordycepin treatment did not affect total Ets-1 expression levels.” These issues are inadequately investigated in the paper as commented on before. Moreover, it is unclear what is meant by “significant” in this passage as no statistics were performed and only one, rather poor looking set of western data is shown. A term like “consistent” or “reproducible” would describe the data if good westerns were available.

Author Response

We have also addressed the concerns and comments raised by you as listed below:

Key issues

Point 1. P13-14: Figure 4 is a key figure with several issues. (It is our understanding from the context and page numbers that the reviewer is not actually referring to Figure 4 but to Figure 5)

Response 1.

We experimented whether cordycepin affected ERK and AKT activation as shown in the figure below. Based on these data, it was concluded that cordycepin inhibited AKT activation while not affecting ERK activation by three independent experiments.

In the first submitted result, pERK1/2 appeared lower in cordycepin-treated T24R2 cells than in untreated ones, but the band intensity of total ERK also was lower in cordycepin-treated cells. In revised result, the intensities of total ERK in untreated and cordycepin-treated T24R2 were similar, and the level of pERK was also similiar in the experimental groups. Therefore, we suggested that cordycepin inhibited the AKT pathway but not ERK in T24R2 cells.

Point 2. 1) the figure is inadequately described in the text. All results need a mention.

Response 2: Thank you for this point. We strengthened the explanation of Fig 5C and Fig 5D as follows: “As shown in Figure 5C, cordycepin treatment decreased the level of phospho-Ets-1(Thr38), which was followed by a reduction of MDR1 expression. Although the levels of activated ERK and AKT were higher in T24R2 cells than in T24 cells, cordycepin treatment had little effect on ERK activation but significantly reduced AKT activation in T24R2 cells (Figure 5D).”

Point 3. 2) it appears from the band shape that the total and phospho-ERK data were not taken from the same blot. There is no loading control for either blot. A loading control should be shown for every blot.

Response 3: Thank you for this observation. In compliance with your request, we replaced Figure 5D with the third panel (3) shown in the figure above.

Point 4. 3) the pETS1Thr38 site is not a normal Akt target site, but thought to be controlled by ERK. In this light, the data in Figure 5D really need to be improved and westerns for ERK and pERK should also be shown for the wortmannin experiment, as crosstalk between the PI3K and ERK pathways could be occuring.

Response 4: Thank you for this precise point. As you mention, we considered any potential specific sequence that may be phosphorylated by AKT using a phophorylation prediction software program. However, phosphorylation of Ets-1 by ERK was well known. Thus, if the activation of ERK were regulated by cordycepin in Fig 5D, phosphorylation of ERK would be seen in Figure 5E, and the ERK-inhibitor-treated group would be also added in Figure 5E and 5F. However, we focused on AKT because we concluded that cordycepin did not affect ERK activation. When we treated cells with wortmannin, a PI3K inhibitor, AKT activation was also inhibited, as was Ets-1 phosphorylation and MDR1 expression, like cordycepin. In addition, as shown in Figure 5F, the sensitivity of T24R2 to cisplatin was increased through the inhibition of PI3K/AKT by wortmannin. Therefore, we concluded that cordycepin inhibited MDR1 expression by inhibiting the PI3K/AKT pathway in T24R2 cells. Nevertheless, as you point out, that specific experimental result has not been performed. However, given that the time constraints for completing this revision (10 days) are too narrow to facilitate completion of a new experiment, we believe that the above explanation provides a reasonable rationale for our conclusion, with the caveat that future experiments must be performed to confirm its validity. We desperately hope that you will find this to be sufficient for the scope of the current manuscript.

Minor issues

Point 1. A) It is unclear to me which comparisons are tested in Figure 1B. Is untreated compared to the increased cordycepin concentration or is the difference between with and without cisplatin tested?

  1. B) Was a correction for multiple measurements, eg Dunnett or Bonferroni used?

Response: a) T24R2 cells showed high resistance to 2 μg/mL cisplatin (Fig. 1A) without cordycepin treatment. A high concentration of cordycepin (50 μg/mL) induced about 20% cell death in T24R2 cells, but 30 μg/mL cordycepin used in our experiments did not induce cell death in T24R2 cells. The sensitivity of T24R2 cells to cisplatin was dependent on the concentration of cordycepin combined with 2 μg/mL cisplatin.

Response: b) We performed Bonferroni correction for multiple comparisons using GraphPad Prism5. The legend of Figure 1 now includes the following sentence: “Bonferroni post-hoc correction for multiple comparisons was performed to compare means by row.”

Point 2. The statements should be rendered more tentative.

- P13 “Taken together, inhibition of AKT activation by cordycepin attenuates Ets-1 activity, which leads to resensitization of T24R2 to cisplatin through the reduction of pEts-1.”

Response 2:  The sentence was replaced with the following: “These data suggest that inhibition of PI3K/AKT activation by cordycepin attenuates Ets-1 activity, which leads to resensitization of T24R2 cells to cisplatin through the reduction of phospho-Ets-1.”

Point 3. And p 15 “Therefore, cordycepin inhibits MDR1 expression via inhibition of Ets-1 activity.”

Response 3: We modified the Discussion section, and this sentence was deleted.

Point 4. P16 “Active PI3K/AKT pathway induces transcriptional upregulation of MDR1 through activation of Ets-1”. These statements are far too definitive.

Response 4: This sentence was replaced with the following, now at p15: “We suggest that codycepin inhibits activation of Ets-1 through inhibition of PI3K/AKT pathway, but not ERK1/2 pathway.”

Point 5. P15 “cordycepin inhibited MDR1 expression at the transcriptional level” To make this statement, there should also be measurement of endogenous MDR1 mRNA levels at the very least and preferably also of unspliced or chromatin bound MDR1 mRNA. This should be not too hard to do with qPCR. The methods suggest that qPCR for MDR1 mRNA was performed, but I don’t see the data.

Response 5: Thank you for these excellent points. The sentence, “cordycepin inhibited MDR1 expression at the transcriptional level,” was deleted when Discussion section was modified. In the Materials and Methods section, “real-time RT-PCR” was replaced with “RT-PCR.”

Point 6. P16 “Various growth factors, …, increase the mRNA level of Ets-1 through the Ras/Raf/MEK/ERK1/2 pathway” and “ Among the modifications of Ets-1, Thr-38 phosphorylation of Est-1 by ERK1/2 was well defined [27-29]. Strangely, no significant decrease in ERK activation was observed in T24R2 treated with cordycepin, while a significant inhibition in PI3K-mediated AKT activation was observed. PI3K/AKT inhibition using wortmannin significantly inhibited the phosphorylation of Ets-1 and MDR1 expression in T24R2 (Figure 5E), and also resensitized T24R2 to cisplatin (Figure 5F). Although it was known that AKT can affect Ets-1 expression levels [30], cordycepin treatment did not affect total Ets-1 expression levels.” These issues are inadequately investigated in the paper as commented on before. Moreover, it is unclear what is meant by “significant” in this passage as no statistics were performed and only one, rather poor looking set of western data is shown. A term like “consistent” or “reproducible” would describe the data if good westerns were available.

Response 6: In the newly written Discussion section (now p 14), the sentence, “Various growth factors, such as hepatocyte growth factor, basic fibroblast growth factor, and vascular endothelial growth factor, increase the mRNA level of Ets-1 through the Ras/Raf/MEK/ERK1/2 pathway.” was replaced with the following: “Transcription and post-translational modification regulate Ets-1 expression and its activity [22,23].”

The sentences, “Among the modifications of Ets-1, Thr-38 phosphorylation of Est-1 by ERK1/2 was well defined [27-29]. Strangely, no significant decrease in ERK activation was observed in T24R2 treated with cordycepin, while a significant inhibition in PI3K-mediated AKT activation was observed. PI3K/AKT inhibition using wortmannin significantly inhibited the phosphorylation of Ets-1 and MDR1 expression in T24R2 (Figure 5E), and also resensitized T24R2 to cisplatin (Figure 5F). Although it was known that AKT can affect Ets-1 expression levels [30], cordycepin treatment did not affect total Ets-1 expression levels.” were replaced with the following: “Several reports have demonstrated that the MEK/ERK signaling pathway regulates Ets-1 expression and its activity through phosphorylation of Ets-1 at Thr 38 [16,24-27]. Strangely, a decrease of ERK activation was not observed in T24R2 cells treated with cordycepin, while the inhibition of AKT activation was observed. Previous studies have reported that AKT activation may induce Ets1 expression [28-30]. In our study, no decrease of Ets-1 expression was observed, whereas phospho-Ets-1 (Thr38) was significantly decreased in T24R2 cells treated with cordycepin or wortmanin. These data suggest that PI3K/AKT inhibition mediated by cordycepin inhibited the phosphorylation of Ets-1 and MDR1 expression in T24R2 cells and resensitized the cells to cisplatin (Fig. 5).”

Reviewer 2 Report

Oh et al. have addressed a very important problem in the cancer field as it is the resistance that certain tumors create against cisplatin.  In this research , the authors have revealed in an elegant way that cordycepin mediates resensitization to cisplatin in T24R2 cells, by controlling AKTT308 phosphorylation,  Ets1 transciptional activity and MDR1 expression. However, there are a number of concerns that authors should address before the manuscript be accepted by IJMS.

Major points:

1- The authors use wortmannin to inhibit AKT, this is a mistake, since wortmaninn is a PI3K inhibitor. when this enzyme is pharmacologically inhibited all its effector molecules are also inhibited. So the conclusion issued by the authors is also an error, the inhibition of Akt phosphorylation does not rescue the sensitivity to cisplatin in T24R2 cells, in this case is PI3 K- dependent. In order to verify the role of AKT in this process, the authors should block the AKT activity using specific inhibitors and on the other hand also investigate whether the resensitization process here described it could involve the PDK1 /nPKCs pathway. This apporach could be made using specific inhibitors for PDK1 and nPKCs.

2- The discussion would be reviewed because, as presented, it seems a repetition of the results.

Minor points:

1- in the Figure 1B, authors have to indicate  the cisplatin concentration used.

2- Regarding Figure 2D, it is not very well expalined, and it is not possible to read what is written inside the boxes.

3- The authors should define the T24 cells  and whenever they refer to them either as T24 or T24R2 they should end or as cell line or cells.

4- To check the English

Author Response

Dear Reviewer:

Thank you very much for permitting us to provide additional information during the review process of our manuscript (Manuscript ID ijms-713488) entitled “Cordycepin resensitizes T24R2 cells, cisplatin-resistant human bladder cancer cell, to cisplatin by inhibiting the AKT-mediated Ets-1 activation.” We appreciate the reviewer’s thoughtful comments and advice.

First, we have changed the title to take into account the reviewers’ comments: “Cordycepin resensitizes T24R2 cisplatin-resistant human bladder cancer cells to cisplatin by inhibiting PI3K/AKT-mediated Ets-1 activation.

We have also addressed the concerns and comments raised by you as listed below:

Major points

Point 1- The authors use wortmannin to inhibit AKT, this is a mistake, since wortmaninn is a PI3K inhibitor. when this enzyme is pharmacologically inhibited all its effector molecules are also inhibited. So the conclusion issued by the authors is also an error, the inhibition of Akt phosphorylation does not rescue the sensitivity to cisplatin in T24R2 cells, in this case is PI3 K- dependent. In order to verify the role of AKT in this process, the authors should block the AKT activity using specific inhibitors and on the other hand also investigate whether the resensitization process here described it could involve the PDK1 /nPKCs pathway. This apporach could be made using specific inhibitors for PDK1 and nPKCs.

Response 1: Thank you for critical and exact points. As your comments say, we cannot claim conclusively that only AKT inhibition resensitizes T24R2 cells to cisplatin. Furthermore, Thus, we revised the manuscript to suggest that inhibition of the PI3K/AKT pathway is a mechanism to overcome the resistance of T24R2 to cisplatin. Experiments using specific inhibitors against AKT, PDK, or nPKCs would be of considerable value, but as our revision must be completed within 10 days, insufficient time remains to purchase materials and design and conduct these experiments.. Therefore, we would like to ask your consideration to allow us to perform such experiments in future projects outside the scope of the current manuscript. I believe your comments will be the driving force for this future project.

Point 2. The discussion would be reviewed because, as presented, it seems a repetition of the results.

Response 2: Thank you for this observation. We have drastically revised the discussion section as a result and feel that your comment has greatly improved our manuscript.

Minor points:

Point 1. in the Figure 1B, authors have to indicate the cisplatin concentration used.

Response 1: We added the concentration of cisplatin in Figure 1B.

Point 2. Regarding Figure 2D, it is not very well explained, and it is not possible to read what is written inside the boxes.

Response 2: We have revised the text in the box to be larger and more legible.

Point 3. The authors should define the T24 cells and whenever they refer to them either as T24 or T24R2 they should end or as cell line or cells.

Response 3: Your comments have been reflected throughout the entire manuscript.

Point 4. To check the English

Response 4: An authorized English editing company (enago) has edited this manuscript.

Round 2

Reviewer 1 Report

The discussion of the paper in particular is much improved and the new western data in Fig 5D are better. I agree with the authors that 10 days for a major revision is far too short, this journal policy is not good for the quality of the science or the usefulness the reviewer’s work. It is frustrating when suggestions that are meant to be helpful in improving a paper could lead to the rejection of a paper, simply because of a ridiculous resubmission deadline. Therefore, I sympathise with their difficulties in providing additional controls. However, there are still a number of issues that really need to be addressed in my opinion.

The title and sentences in the abstract are still too certain for the quality of the evidence for a causal link between PI3K/Akt signalling, ETS phosphorylation and activation, and MDR1 expression presented in this paper. For this far more molecular detail would be required. The statements need to be further qualified. Eg a better title would be “Cordycepin re-sensitizes T24R2 cisplatin-resistant bladder cancer cells by inactivating ETS-1 dependent MDR1 transcription” . This sentence in the abstract: “Cordycepin decreased pThr-38 Ets-1 levels through inhibition of PI3K/AKT, which reduced MDR1 transcription and induced the resensitization of T24R2 cells to cisplatin.” is also not sufficiently supported by the evidence. Better would be for instance “Cordycepin decreased pThr-38 Ets-1 levels and reduced MDR1 transcription, probably through its effects on PI3K/AKT signalling, inducing the resensitization of T24R2 cells to cisplatin.”

The effect of cordycepin on PI3K/Akt signalling is agreed upon by nearly all papers on the subject and therefore needs no further strengthening of the evidence. However, the effect of cordycepin on the ERK pathway is more controversial, with some papers reporting no effect and others an increase. However in some cell types a decrease in phospho-Erk is reported, eg

https://www.ncbi.nlm.nih.gov/pubmed/30410556

https://www.ncbi.nlm.nih.gov/pubmed/30124145

https://www.ncbi.nlm.nih.gov/pubmed/27225448

It is therefore important to have sufficient evidence before it can be stated that ERK is not affected by cordycepin treatment. If the total level of ERK goes down, with the p-ERK going down in proportion, then this will also reduce ERK signalling. The new panel looks much better and it is very reassuring to see similar results for the other proteins in this panel also, but it is now only one set of data in with the original data suggesting a reduction in ERK levels. The authors could have strengthened their case by showing all the replicates of the western blots from Fig 5D in a supplement, this would not have required additional experiments, as they state in the response to my review that the experiment was done in independent triplicates. I still cannot find the statement that the western blots were done in independent triplicates in the paper, this needs to be stated explicitly in the legend of each figure containing western data and/or in the methods.

As indicated in my previous review, it is not convincing to use loading controls from different gels than the ones probed for the query protein, as is clearly the case in many western data in this paper. Eg, if there were appropriate loading controls of the ERK and pERK blots of the original Fig 5D, showing that the last two lanes were underloaded, this would have strengthened the case that ERK is not affected. If at all possible, provide more loading controls where different gels were used.

In the absence of higher quality evidence, the authors should withdraw or soften their assertion that ERK is not involved in the phosphorylation of ETS in this case.

The editing has improved the English of text in several places, but not always. Eg “apoptotic machinery in cancer cells, a small subset of them can acquire resistance” & the deletion of “The” in the first sentence of the results should be reversed, or replaced by “An”. Also in the introduction: “Although cisplatin induced T24 cell death that increased” etc.

“Figure 1. Effects of cordycepin and/or cisplatin treatment on T24R2-cell sensitivity to cisplatin”

Fig 1C misspelling of Cordycepin

“Bonferroni post-hoc correction for multiple comparisons was performed to compare means by row” – Thanks to the authors for adding the information on the Bonferroni correction. However, this does still not tell the reader which comparisons were made, as it is dependent on how the numbers were entered in the spreadsheet. Eg in 1B, are you comparing the effect of cisplatin at the same concentration of cordycepin, or are you comparing the effect of cordycepin to the matched control (with or without cisplatin). You can add brackets to show the comparisons tested.

Author Response

Dear Reviewer:

Thank you very much for permitting us to provide additional information during the review of our manuscript (Manuscript ID ijms-713488) entitled “Cordycepin resensitizes T24R2 cisplatin-resistant human bladder cancer cells to cisplatin by inhibiting PI3K/AKT-mediated Ets-1 activation.” We appreciate the reviewer’s thoughtful comments and advice.

First, we have changed the title to take into account your comment: “Cordycepin Resensitizes T24R2 Cisplatin-Resistant Human Bladder Cancer Cells to Cisplatin by Inactivating Ets-1 dependent MDR1 transcription

We have also addressed the concerns and comments raised by you as listed below:

Point 1. This sentence in the abstract: “Cordycepin decreased pThr-38 Ets-1 levels through inhibition of PI3K/AKT, which reduced MDR1 transcription and induced the resensitization of T24R2 cells to cisplatin.” is also not sufficiently supported by the evidence. Better would be for instance “Cordycepin decreased pThr-38 Ets-1 levels and reduced MDR1 transcription, probably through its effects on PI3K/AKT signalling, inducing the resensitization of T24R2 cells to cisplatin.” 

Response 1.

Thank you for your kindness. As your advice, we changed the sentence with “Cordycepin decreased pThr-38 Ets-1 levels and reduced MDR1 transcription, probably through its effects on PI3K signaling, inducing the resensitization of T24R2 cells to cisplatin.”. Another reviewer advised to change PI3K/AKT to PI3K and we changed PI3K/AKT with PI3K according to the advice in this revised manuscript.

Point 2. 1) The effect of cordycepin on PI3K/Akt signalling is agreed upon by nearly all papers on the subject and therefore needs no further strengthening of the evidence. However, the effect of cordycepin on the ERK pathway is more controversial, with some papers reporting no effect and others an increase. However in some cell types a decrease in phospho-Erk is reported, eg

https://www.ncbi.nlm.nih.gov/pubmed/30410556

https://www.ncbi.nlm.nih.gov/pubmed/30124145

https://www.ncbi.nlm.nih.gov/pubmed/27225448

Response 2: We appreciate your comment for our discussion. We added the following sentence in discussion part and references you introduced.

“ Some reports suggested that cordycepin inhibits ERK activation in some murine cells, including precursors of osteoblast and osteoclast and adipocytes, and human hepatocarcinoma cells [29-31]. Strangely, a decrease in ERK activation was unclear or little in T24R2 cells treated with cordycepin, whereas the inhibition of AKT activation was observed.”

  1. Yu, S.B.; Kim, H.J.; Kang, H.M.; Park, B.S.; Lee, J.H.; Kim, I.R. Cordycepin Accelerates Osteoblast Mineralization and Attenuates Osteoclast Differentiation In Vitro. Evidence-based complementary and alternative medicine : eCAM 2018, 2018, 5892957, doi:10.1155/2018/5892957.
  2. Li, Y.; Wang, X.; Xu, H.; Wang, C.; An, Y.; Luan, W.; Li, S.; Ma, F.; Ni, L.; Liu, M., et al. Cordycepin Modulates Body Weight by Reducing Prolactin Via an Adenosine A1 Receptor. Current pharmaceutical design 2018, 24, 3240-3249, doi:10.2174/1381612824666180820144917.
  3. Zhou, Y.; Guo, Z.; Meng, Q.; Lu, J.; Wang, N.; Liu, H.; Liang, Q.; Quan, Y.; Wang, D.; Xie, J. Cordycepin Affects Multiple Apoptotic Pathways to Mediate Hepatocellular Carcinoma Cell Death. Anti-cancer agents in medicinal chemistry 2017, 17, 143-149.

Point 3. The authors could have strengthened their case by showing all the replicates of the western blots from Fig 5D in a supplement, this would not have required additional experiments, as they state in the response to my review that the experiment was done in independent triplicates. I still cannot find the statement that the western blots were done in independent triplicates in the paper, this needs to be stated explicitly in the legend of each figure containing western data and/or in the methods.

Response 3: As your recommendation, we added a supplementary material showing another western blot data. We tried to calculate the ratios of phospho-form to total form in the material. The data contained in this manuscript is representative results from 2 or 3 independent experiments. We added the following sentence in each figure legend. “The results are representative of at least two independent experiments. ”

Point 4. In the absence of higher quality evidence, the authors should withdraw or soften their assertion that ERK is not involved in the phosphorylation of ETS in this case.

Response 4: Acccording to your advice, the sentence, “We suggest that codycepin inhibits activation of Ets-1 through inhibition of the PI3K pathway, but not the ERK1/2 pathway”, was revised. → “We suggest that codycepin inhibits activation of Ets-1 through inhibition of the PI3K pathway”. 

Point 5. Editing issues

Response 5.

  • “apoptotic machinery in cancer cells, a small subset can acquire resistance” in introduction session → “the apoptotic machinery in cancer cells, a small subset can acquire resistance”

  1. “MTT assay” in the first sentence of the results → The MTT assay
  2. In result 2.1., “Although cisplatin induced T24 cell death that increased in concentration-dependent manner, no significant effect was observed in T24R2 cells, which displayed clear resistance to cisplatin.” → “Although cisplatin induced concentration-dependent T24 cell death, no significant effect was observed in T24R2 cells, which showed clear resistance to cisplatin (Figure 1A).”
  3. “Figure 1. Effects of cordycepin and/or cisplatin treatment on T24R2-cell sensitivity to cisplatin” → revised

Fig 1C misspelling of Cordycepin → revised

Point 6. “Bonferroni post-hoc correction for multiple comparisons was performed to compare means by row” – Thanks to the authors for adding the information on the Bonferroni correction. However, this does still not tell the reader which comparisons were made, as it is dependent on how the numbers were entered in the spreadsheet. Eg in 1B, are you comparing the effect of cisplatin at the same concentration of cordycepin, or are you comparing the effect of cordycepin to the matched control (with or without cisplatin). You can add brackets to show the comparisons tested.

Response 6. Thank you for your advice. The sentence,(the effect of cordycepin was compared in matched group in the presence or absence of cisplatin)”, was added in Figure 1’s legend.

Reviewer 2 Report

The authors do not demonstrate the relationship between AKT and ETS1, they inhibite PI3K  by using wortmannin, consequently, authors relate PI3k with ETS1. 

It is imperative  to change PI3K/AKT pathway or PI3K/AKT activation by PI3K pathway or PI3k activation, in the title and in the manuscript.

The rest of the remarksthat I pointed out in the first review the authors have corrected them.

Author Response

Dear Reviewer:

Thank you very much for permitting us to provide additional information during the review of our manuscript (Manuscript ID ijms-713488) entitled “Cordycepin resensitizes T24R2 cisplatin-resistant human bladder cancer cells to cisplatin by inhibiting PI3K/AKT-mediated Ets-1 activation.” We appreciate the reviewer’s thoughtful comments and advice.

Point: The authors do not demonstrate the relationship between AKT and ETS1, they inhibite PI3K  by using wortmannin, consequently, authors relate PI3k with ETS1. 

It is imperative to change PI3K/AKT pathway or PI3K/AKT activation by PI3K pathway or PI3k activation, in the title and in the manuscript.

Response: Thank you for your kind advice. We agree with your recommendation. PI3k/AKT was replaced with PI3K in this revised manuscript.

Additional Information: We changed the title according to another reviewer’s comment. “Cordycepin Resensitizes T24R2 Cisplatin-Resistant Human Bladder Cancer Cells to Cisplatin by Inactivating Ets-1 dependent MDR1 transcription”